# Comparative Genomic Analysis of a Novel *Vibrio* sp. Isolated from an Ulcer Disease Event in Atlantic Salmon (*Salmo salar*)

**DOI:** 10.3390/microorganisms11071736

**Published:** 2023-07-02

**Authors:** Maryam Ghasemieshkaftaki, Ignacio Vasquez, Aria Eshraghi, Anthony Kurt Gamperl, Javier Santander

**Affiliations:** 1Marine Microbial Pathogenesis and Vaccinology Laboratory, Department of Ocean Sciences, Memorial University of Newfoundland, St. John’s, NL A1C 5S7, Canada; mghasemieshk@mun.ca (M.G.); ivasquezsoli@mun.ca (I.V.); 2Department of Infectious Diseases & Immunology, University of Florida, Gainesville, FL 32608, USA; ariaeshraghi@ufl.edu; 3Fish Physiology Laboratory, Department of Ocean Sciences, Memorial University of Newfoundland, St. John’s, NL A1C 5S7, Canada; kgamperl@mun.ca

**Keywords:** Atlantic salmon, *Vibrio* sp. J383, genomics, phylogenetics, phenotype, ulcer disease

## Abstract

Ulcer diseases are a recalcitrant issue at Atlantic salmon (*Salmo salar*) aquaculture cage-sites across the North Atlantic region. Classical ulcerative outbreaks (also called winter ulcer disease) refer to a skin infection caused by *Moritella viscosa*. However, several bacterial species are frequently isolated from ulcer disease events, and it is unclear if other undescribed pathogens are implicated in ulcer disease in Atlantic salmon. Although different polyvalent vaccines are used against *M. viscosa*, ulcerative outbreaks are continuously reported in Atlantic salmon in Canada. This study analyzed the phenotypical and genomic characteristics of *Vibrio* sp. J383 isolated from internal organs of vaccinated farmed Atlantic salmon displaying clinical signs of ulcer disease. Infection assays conducted on vaccinated farmed Atlantic salmon and revealed that *Vibrio* sp. J383 causes a low level of mortalities when administered intracelomic at doses ranging from 10^7^–10^8^ CFU/dose. *Vibrio* sp. J383 persisted in the blood of infected fish for at least 8 weeks at 10 and 12 °C. Clinical signs of this disease were greatest 12 °C, but no mortality and bacteremia were observed at 16 °C. The *Vibrio* sp. J383 genome (5,902,734 bp) has two chromosomes of 3,633,265 bp and 2,068,312 bp, respectively, and one large plasmid of 201,166 bp. Phylogenetic and comparative analyses indicated that *Vibrio* sp. J383 is related to *V. splendidus*, with 93% identity. Furthermore, the phenotypic analysis showed that there were significant differences between *Vibrio* sp. J383 and other *Vibrio* spp, suggesting J383 is a novel *Vibrio* species adapted to cold temperatures.

## 1. Introduction

Ulcerative diseases in Atlantic salmon (*Salmo salar*) aquaculture were first reported in Norway in 1980, and still are a significant health/economic issue for the North Atlantic region [1]. The Gram-negative marine pathogen *Moritella viscosa* is typically described as the etiological agent of the classic ulcer disease (also called winter ulcer disease) in European farmed fish [2,3]. Although *M. viscosa* is the primary pathogen associated with ulcer disease, it is not the only bacterium isolated from ulcers and lesions in Atlantic salmon. The isolation and identification of various bacterial species from ulcer-disease cases have been reported, including *Aliivibrio wodanis* (formerly *Vibrio wodanis*) and *Tenacibaculum* sp. [4,5]. In fact, *Tenacibaculum* might target scarified skin, and co-infect wounds with *M. viscosa* [5]. Also, *M. viscosa* might co-infect with *A*. *wodanis*, which may limit the growth of *M. viscosa* [6]. In addition, *A. wodanis* has the ability to solo infect Atlantic salmon [7].

Since the first documented outbreak of ulcerative disease in Eastern Canada (summer 1999) caused by *A. wodanis* in Atlantic salmon [8], several pathogens including *M. viscosa* and *Tenacibaculum* spp. have been isolated [5,9]. Despite broad immunization with polyvalent vaccines containing *M. viscosa* antigens, ulcerative-disease events continue to be reported, causing poor fillet quality and financial losses [10,11]. This suggests that undescribed bacterial species may be involved in ulcer disease pathology [11], and that they are not covered by current vaccines. 

The occurrence of ulcerative disease in Norway and other European countries is significantly different from in Canada. In European countries, ulcerative disease occurs below 7 °C [12]. However, in Canada, ulcerative disease frequently occurs in summer and mid-autumn when water temperatures are over 10 °C [13]. The mortality rate caused by ulcerative disease is less than 10% in Norwegian farms during winter outbreaks, and fish that survive recover when the water temperature increases above 8 °C in spring [9]. In Atlantic Canada, the highest cumulative cage-level mortality recorded was 31.2% in the first described outbreak [8]; however, it seems that mortality has decreased over time, although its frequency has increased [13].

Skin-ulcer disease in Atlantic salmon is relatively unexplored, and there is limited published data on this disease in Atlantic salmon on the east coast of Canada [13,14]. Salmon farmers report that skin ulcers in sea-cages can progress very fast. At first, fish may only have laterally raised scales, and after a few days, they die with a single large circular ulcerative lesion of several centimeters in diameter [14]. Actually, in Atlantic Canada, “summer” skin ulcers in Atlantic salmon lead to significant mortality rates and economic losses [13].

In this study, we isolated a novel *Vibrio* sp. J383 strain from vaccinated farmed Atlantic salmon exhibiting clinical signs of skin ulcer disease, and characterized its phenotype and genome. Infection assays revealed that *Vibrio* sp. J383 does not cause an acute infection, but instead causes a chronic infection in Atlantic salmon. Comparative genomics analyses suggest that *Vibrio* sp. J383 is a new species that might contribute to skin ulcer disease in Atlantic salmon.

## 2. Materials and Methods

### 2.1. Phenotypic Characterization

#### 2.1.1. Isolation

Farmed Atlantic salmon vaccinated with ALPHA JECT exhibiting clinical signs of ulcer disease at 12 °C (Figure 1) were netted and immediately euthanized with an overdose of MS-222 (400 mg/L; Syndel Laboratories, BC, Canada). Tissue samples (spleen, head kidney, and liver) were collected and placed into sterile homogenizer bags (Nasco whirl-pak^®^, Fort Atkinson, WI, USA, then weighed and homogenized in phosphate-buffered saline [PBS; 136 mM NaCl, 2.7 mM KCl, 10.1 mM Na_2_HPO_4_, 1.5 mM KH_2_PO_4_ (pH 7.2) up to a final volume of 1 mL] [15]. From the homogenized tissue suspension, 100 μL was plated onto Trypticase Soy Agar (TSA; Difco, Franklin Lakes, NJ, USA) supplemented with up to 2% NaCl and incubated at 15 °C for 48–72 h. Colonies were streak purified on agar for further analysis [16]. Bacterial stocks were preserved at −80 °C in 10% glycerol and 1% peptone solution. A single colony of each isolated bacteria was grown in 3 mL Trypticase Soy Broth (TSB) supplemented with 2% NaCl in a 16 mm diameter glass tube and placed in a drum roller (TC7, New Brunswick Scientific, MA, USA) for 24 h at 15 °C with aeration (180 rpm). When required, TSB was supplemented with 1.5% bacto-agar (Difco) and 0.02% Congo-red (Sigma-Aldrich, Burlington, MA, USA) [17,18]. Luria Bertani (LB; yeast extract 5 g; tryptone 10 g; NaCl 10 g; dextrose 1 g) with different concentrations of NaCl (0, 0.5, 2%) was used to evaluate halophilic growth [19]. 

#### 2.1.2. Biochemical, Enzymatic and Physiological Characterization

The biochemical profile of *Vibrio* sp. J383 was characterized using API 20E, API 20NE, and API ZYM according to the manufacturer’s (BioMerieux, Marcy-l’Etoile, France) instructions. The strips were incubated at 15 °C for 48 h, and the results were analyzed using API web (BioMerieux). The primary characterization of *Vibrio* sp. J383 was performed based on the Gram stain, capsule stain, and morphological and cultural characteristics [16]. *Vibrio* sp. J383 growth rate was determined in TSB with 2% NaCl at 4, 15, 28, and 37 °C. Halophilic growth was evaluated in LB supplemented with 0.5 and 2% of NaCl. Also, catalase and oxidase activity were measured according to standard protocols [16,19]. Hemolytic activity was assessed in TSA with 5% salmon blood and sheep blood agar at 15 °C [19,20]. The bacteria’s liposaccharide (LPS) profile was examined based on previous protocols [21,22]. *Vibrio* sp. J383 growth curves were determined in triplicate at 15 °C according to established protocols [18].

#### 2.1.3. Antibiogram

The susceptibility to antimicrobials was determined using sensi-disc diffusion tests [23]. Briefly, *Vibrio* sp. J383 susceptibility was determined for tetracycline (10 μg), oxytetracycline (30 μg), ampicillin (10 μg), sulfamethoxazole (STX) (25 μg), chloramphenicol (30 μg), colistin sulphate (10 μg), and oxalinic acid (2 μg) using standard methods [19,23].

#### 2.1.4. Siderophore Synthesis

A siderophores secretion assay was performed using CAS plates according to standard procedure [24]. Briefly, previously mentioned conditions were used to cultivate *Vibrio* sp. J383, and mid-log phase bacteria with an optical density (O.D.) at 600 nm of 0.7 were harvested, washed three times with PBS at 6000 rpm for 10 min, and then resuspended in 1 mL of PBS. This bacterial culture was used to inoculate TSB with 2% NaCl, and TSB with 2% NaCl supplemented with 100 μM of FeCl_3_ or 100 μM of 2,2 dipyridyl in a ratio 1:10 (bacteria: culture media). *Vibrio* sp. J383 was cultured under aeration for 24 h at 15 °C. The cells were collected at the mid-log phase after the incubation time, washed twice with PBS, and resuspended in 100 μL of PBS. After that, CAS agar plates were inoculated with 5 μL of the concentrated bacterial pellet and incubated at 15 °C for 48 h [24].

### 2.2. Infection Trials

#### 2.2.1. Fish Origin and Holding Conditions

Farmed Atlantic salmon (~200–250 g) of New Brunswick (Saint John River) origin that had been vaccinated with ALPHA JECT micro IV (Pharmaq, Overhalla, Norway) were held at the Joe Brown Aquatic Research Building (Ocean Sciences Center, Memorial University; MUN) in 3800 L tanks supplied with 95–100% air saturated, and UV-treated, flow through seawater at 10–12 °C, and an ambient photoperiod (spring–summer). The fish were fed three days per week at 1% body weight with a commercial dry pellet (Skretting, BC, Canada; 50% protein, 18% fat, 1.5% carbohydrate, 3% calcium, 1.4% phosphorus). All experiments were conducted under approved institutional animal ethics protocols (#18-1-JS and #18-03-JS).

#### 2.2.2. Bacterial Inoculum Preparation

Isolated strains were grown, harvested, and used to infect the Atlantic salmon. Briefly, bacterial cells were harvested at the mid-log phase, at an O.D. at 600 nm of ~0.7, and washed three times with PBS at 6000 rpm for 10 min. Bacterial O.D. was monitored using a Genesys 10 UV spectrophotometer (Thermo Spectronic, Thermo Fischer Scientific, MA, USA) and by plating to determine the colony forming units (CFU/mL) [25,26].

#### 2.2.3. Koch’s Postulates

The infection procedures were conducted in the AQ2 biocontainment unit at the Cold-Ocean Deep-Sea Research Facility (CDRF), MUN. Fish were transferred to the AQ2-CDRF unit and acclimated for one week at 10 °C before infection. An initial infection screening assay was conducted in Atlantic salmon (200 g) intraperitoneally (ip) injected with a high dose (1 × 10^8^ CFU/dose) of each isolated strain. Each infected group consisted of 5 fish in individual tanks under optimal conditions. Fish ip injected with PBS were used as a negative control, and fish ip injected with *M. viscosa* J311 (ATCC BAA-105) were used as a positive control. 

A total of 135 Atlantic salmon (~250 g) were used to evaluate Koch’s postulates for *Vibrio* sp. J383 [15,26,27]. The infection procedures were conducted according to established protocols [25,28]. Briefly, fish were anesthetized with 0.05 g/L MS-222 and individually injected with 100 μL of the respective bacterial inoculum. Fish were divided into three 500 L tanks containing 45 Atlantic salmon each, and intracelomic (ic) infected with 10^6^, 10^7^, and 10^8^ CFU/dose, respectively. Mortality was monitored until 12 weeks post-infection (wpi). The water temperature was increased during the experiment, starting at 10 °C for 4 wpi, then raised to 12 °C at 5 wpi, and finally increased to 16 °C at 9 wpi until 12 wpi. Tissue samples (e.g., head kidney, liver, spleen, and blood) from 6 fish of each dose were aseptically taken at 2 wpi. Bacterial loads were determined by established protocols [25,28]. Briefly, liver, spleen and head kidney were aseptically removed, and sections of the collected tissues were placed into sterile homogenizer bags (Nasco whirl-pak^®^, Fort Atkinson, WI, USA), weighed, and PBS was added to a final volume of 1 mL. Then, the tissues were homogenized, the suspensions were serially diluted (1:10), and then plated onto TSA supplemented with 2% NaCl. To determine the number of bacteria CFU per g of tissue, the plates were incubated at 15 °C for at least 5 days. The total bacterial count was normalized to 1 g of tissue according to the initial weight of the tissue as previously described [25]. Additionally, blood samples were taken from 6 fish per dose every two weeks until the end of the experiment (i.e., at 2, 4, 6, 8, 10, and 12 wpi). Heparin (100 mg/mL) was added to the blood samples, serially diluted in filtered sterilized seawater (1:10) and plated in TSA with 2% NaCl. 

Since mortality was evident at 12 °C, a second infection trial was conducted only at 12 °C. A total of 160 Atlantic salmon (~250 g) were equally distributed in four 500 L tanks containing 40 fish each and acclimated at 12 °C for one week before infection. One group of fish was not infected and used as a negative control. Fish in the other group were ic injected with 10^8^ CFU/dose of *Vibrio* sp. J383. Blood samples were collected randomly from 9 fish every two weeks to determine the bacterial load until 14 wpi. Also, tissue samples (e.g., head kidney, liver, and spleen) were collected at 12 and 14 wpi from 9 fish. This experiment was conducted according to established protocols [25,28], and mortality was monitored daily until 12 wpi.

### 2.3. Vibrio sp. J383 Genomics

#### 2.3.1. *Vibrio* sp. J383 DNA Extraction and Sequencing

*Vibrio* sp. J383 was grown, harvested, and washed as previously described. According to the manufacturer’s instructions (Promega, Madison, WI, USA), the Wizard Genomic DNA Purification Kit was used to extract the genomic DNA (gDNA) of *Vibrio* sp. J383. The gDNA was quantified by spectrophotometer using a Genova Nano Micro-Spectrophotometer (Jenway, UK) and evaluated for purity and integrity by electrophoresis (0.8% agarose gel) [29]. Libraries and sequencing were conducted at Genome Quebec (Canada) using PacBio and Miseq Illumina sequencers.

#### 2.3.2. Genome Assembly, Annotation and Data Submission

Celera Assembler (August 2013 version) was used to assemble the PacBio readings. The assembled contigs were analyzed using a CLC genomic workbench (Qiagen, v22.0). Genome annotations were conducted using the Rapid Annotation Subsystem Technology pipeline (RAST 2.0.) (http://rast.nmpdr.org/; accessed on 2 February 2023) [30,31], and PATRIC (https://www.patricbrc.org; accessed on 2 February 2023) [32]. The *Vibrio* sp. J383 genome was submitted to the National Center for Biotechnology Information (NCBI) for public accessibility and re-annotated using the NCBI Prokaryotic Genome Annotation Pipeline. The *Vibrio* sp. J383 chromosomes and plasmid genomes were visually mapped using CG view software (https://cgview.ca/ (accessed on 20 June 2023).

#### 2.3.3. Comparative Genomics Analysis

Average nucleotide identity (ANI) was calculated by whole genome alignments using the CLC genomic workbench whole genome analysis tool with default parameters. 

(Min. initial seed length = 15; Allow mismatches = yes; Min. alignment block = 100). A minimum similarity of 0.8 and a minimum length of 0.8 were used as parameters for CDS identity. A comparative heat map was made using CLC. Phylogenetic analysis was performed using two different software packages, CLC Genomic workbench v22.0 and MEGA11 [33]. Evolutionary history was estimated using the Neighbor-Joining method with a bootstrap consensus of 500 replicates [34], and evolutionary distance was computed using the Jukes–Cantor method [35]. *Photobacterium damselae* 91-197 (AP018045/6) chromosomes were utilized as an outgroup [36]. Whole genome dot plots between closely related strains were constructed using the whole genome analysis tool to visualize and analyze genomic differences. Comparative alignment analysis was conducted using the CLC genomic workbench (Qiagen, v22.0). This analysis was used to identify homologous regions (locally collinear blocks), translocations, and inversions within the two bacterial genomes for chromosomes 1 and 2. 

#### 2.3.4. Genomic Islands

The detection of genomic islands (GIs) was conducted using the Island Viewer v.4 pipeline (https://www.pathogenomics.sfu.ca/islandviewer/browse/; accessed on 1 February 2023), which integrates Island Path-DIMOB, SIGH-HMM, and Island Pick analysis tools into a single platform [37]. Analysis was performed for both chromosomes and the plasmid. SecReT6 v3 web server was utilized to identify and annotate the type VI secretion system (T6SS) genes that share sequence homology with characterized T6SSs [38,39].

### 2.4. Statistical Analysis

Fish survival rates were transformed using an arc-sin (survival rate ratio) function. One-way ANOVAs were utilized to identify significant differences. GraphPad Prism 9 was used to conduct all statistical analyses (GraphPad Software, California, CA, USA). 

## 3. Results

### 3.1. Phenotypic Characterization

*Vibrio* sp. J383 displayed substantial growth in TSB with 2% NaCl between 15 °C and 4 °C (Table 1). However, it did not grow well at 28 °C and did not grow at 37 °C. *Vibrio* sp. J383 grew well in LB with 1% and 2% NaCl at 15 °C. However, it did not grow in LB supplemented with 0 and 0.5% NaCl at 15 °C. These results indicated that *Vibrio* sp. J383 is psychotropic and halophilic. *Vibrio* sp. J383 showed hemolytic activity in sheep and salmon blood agar at 15 °C (Appendix A). *Vibrio* sp. J383 was shown to be motile, oxidase and catalase-positive, and type I-fimbria-negative (Table 1).

To evaluate siderophore synthesis, *Vibrio* sp. J383 was grown under iron-enriched and iron-limited conditions, inoculated onto CAS agar plates, and incubated at 15 °C for 48 h. Siderophore secretion was observed under iron-enriched (100 μM of FeCl_3_), iron-limited (100 μM of 2,2 dipyridyl), and control conditions (TSB). Furthermore, there were no noticeable differences in the size of the halo for siderophore secretion between different groups (Appendix A).

*Vibrio* sp. J383 produces a capsule and a smooth lipopolysaccharide (LPS) profile (Appendix A).

The biochemical and enzymatic profiles indicated that *Vibrio* sp. J383 can synthesize alkaline phosphatase, esterase (C4), esterase lipase (C8), lipase (C14), leucine, valine, and cysteine arylamidase, acid phosphatase, naphthol-AS-BI phosphohydrolase, and galactosidase (Appendix A). *Vibrio* sp. J383 reduces nitrates and glucose, produces indole from tryptophan, produces esculinase and gelatinase, β-galactosidase and can utilize D-mannitol, D-glucose and D-amygdaline (Appendix A). The API 20NE profile 7474004 indicated that the isolate could be *V. vulnificus* or *V. alginolyticus* with 64.8% and 34.6% confidence, respectively (Appendix A).

### 3.2. Antibiogram

The antibiogram analysis showed that *Vibrio* sp. J383 is colistin-sulphate-resistant, but susceptible to ampicillin, tetracycline, oxytetracycline, sulfamethoxazole, chloramphenicol, oxalinic acid, and the vibriostatic agent O-129 (Table 1). These results are similar to other *Vibrio* spp. strains [40].

### 3.3. Infection Trials in Atlantic Salmon

We isolated a total of five strains, three from the head kidney, one from the liver, and one from the spleen, from different infected fish. An initial screening to determine the virulence of the five isolates was conducted in Atlantic salmon (200 g). The fish were transferred to the AQ2/3 biocontainment zone of the CDRF, acclimated for 1 week, and intraperitoneally (ip) injected with a very high dose (1 × 10^8^ CFU/dose) of each isolate. Fish ip injected with PBS were used as a negative control, and fish ip injected with *M. viscosa* J311 were used as a positive control. As expected, *M. viscosa* killed all the animals quickly, and all the animals ip injected with PBS survived (Appendix A). Only the strain J383 (SP6) caused mortality and clinical signs of ulcer disease. Mortality associated with J383 infection in Atlantic salmon indicates that it causes a chronic type of infection rather than an acute infection. This is consistent with the infection event from which these samples were obtained. 

The first infection trial was conducted to evaluate Koch’s postulates under rising temperature conditions (10 °C for 4 weeks, 12 °C for 4 weeks, and 16 °C for 4 weeks). Approximately 22.5% mortality was recorded in the high-dose (10^8^ CFU/dose) group, 5% mortality rate in the medium-dose (10^7^ CFU/dose) group, and no mortality was observed in the low-dose (10^6^ CFU/dose) group (Figure 2A). Mortality started at 10 °C, peaked at 12 °C, but was not reported at 16 °C. Clinical signs and presence of *Vibrio* sp. J383 were observed in all moribund Atlantic salmon (Figure 2B). *Vibrio* sp. J383 was not detected in internal organs and blood samples at 2 wpi, but it was detected in blood in the high-dose infection group at 4, 6, 8 and 10 wpi. *Vibrio* sp. J383 was detected in the blood of fish infected with the medium dose at 6 and 10 wpi. However, at 12 wpi, no bacteria were observed in the collected samples from the different doses (Table 2). The isolation of *Vibrio* sp. J383 from blood samples confirmed Koch’s postulates.

A second infection assay was performed at 12 °C to determine the infection kinetics of *Vibrio* sp. J383, with 10^8^ CFU/dose. Mortality started at 5 dpi and reached 20% by 90 dpi (Figure 3). *Vibrio* sp. J383 caused bacteremia in about 30–100% of the infected fish, and reached a peak at 6 wpi, which is consistent with the mortality levels (Figure 3 and Figure 4A). Bacteremia started to decrease by 8–10 wpi. No bacteria in the blood were detected after 12 wpi (Figure 4A), but *Vibrio* sp. J383 was detected in the spleen, liver, and head kidney at this sampling point (Figure 4B). Higher bacterial loads were observed in the spleen samples compared to the liver and head kidney (Figure 4B). 

### 3.4. Vibrio sp. J383 Genomics

*Vibrio* sp. J383 gDNA sequenced by PacBio and MiSeq revealed the presence of two chromosomes and one large plasmid. *Vibrio* sp. J383 chromosome 1 (NZ_CP097293.1) has 3,633,265 bp, chromosome 2 (NZ_CP097294.1) 2,068,312 bp, and the large plasmid pJ383 (NZ_CP097295.1) 201,166 bp (Figure 5A–C). The coverage assembly for chromosome 1, chromosome 2, and the parge plasmid was 306, 27, and 14 times, respectively. The plasmid profile agrees with the genomic analysis, supporting the theory that this *Vibrio* sp. possesses one large plasmid and no small plasmids. *Vibrio* sp. J383′s genome was submitted to NCBI under the BioProject (PRJNA836625) and BioSample (SAMN28165975). *Vibrio* sp. J383 genome has a total estimated length of 5.9 Mb and a G + C content of 44.3 and 44.1% for chromosomes 1 and 2, respectively. RAST pipeline annotation predicted a total of 309 subsystems and 3235 coding sequences (CDS) for chromosome 1, a total of 101 subsystems and 1866 CDSs for chromosome 2, and a total of 8 subsystems and 237 CDSs for the large plasmid p.J383 (Table 3). The NCBI Prokaryote Genome Annotation pipeline (PGAP) presented a total of 5288 genes predicted, a total of 16 (5S), 15 (16S), and 15 (23S) rRNAs, 138 tRNAs, and 5 ncRNAs for the whole genome (Table 4).

### 3.5. Genomic Islands (GIS)

Twenty-four putative GIs were identified within the chromosomes and plasmid: sixteen GIs in chromosome 1, seven GIs in chromosome 2, and one GI in the plasmid (Figure 6A–C). The GIs’ size ranged from 6 kb to 50 kb, with a total of 929 genes (Appendix A). Genes encoding for integrases, transposases, phage integrase, and multidrug-resistance transporters were found in GIs 6, 7, 9 and 12, respectively. Also, UDP-3-O-[3-hydroxymyristoyl] glucosamine N-acyltransferase, virulence factor VirK and alpha-galactosidase were in GIs 3, 7 and 15, respectively. Zonula occludes toxin (Zot)-like phage protein was found in the chromosome 1 of *Vibrio* sp. J383. Glutaredoxin encoding gene was detected in genomic island 20 of chromosome 2. 

The type VI secretion system (T6SS) is an important virulence factor detected in *Vibrio* sp. J383. The SecReT6 v3 web server identified and annotated the T6SS genes that share sequence homology with characterized T6SSs and indicated that the *Vibrio* sp. J383 genome encodes two distinct T6SSs in GI 12 of chromosome 1 and GI 21 of chromosome 2 (Table 5, Figure 7, and Appendix A). The *Vibrio* sp. J383 secretion systems belong to the T6SS*^i^* family and are most closely related to the T6SS in *Vibrio coralliilyticus* OCN008. Some genes encoded within these two loci do not share sequence homology with known T6SS genes; however, predicting the structures of the encoded proteins with AlphaFold and comparison to solved structures in the Protein Data Bank (http://rcsb.org) (last accessed on 20 June 2023) allowed us to identify additional T6SS genes [41]. These analyses indicate that both T6SS loci in the *Vibrio* sp. J383′s genome encode for the full complement of components required for T6SS assembly and activity [42].

### 3.6. Comparative Genomic Analysis

The phylogenetic analysis of *Vibrio* sp. J383 chromosomes indicated that it was closely related to *V. splendidus* (Figure 8A–D). The average nucleotide identity (ANI) analysis between *Vibrio* sp. J383 and *V. splendidus* showed 95.75% identity for chromosome 1 (Appendix A) and 93.31% identity for chromosome 2 (Appendix A), which suggests that these two strains share a common ancestor. 

However, the dot plot showed significant differences between *Vibrio* sp. J383 and *V. splendidus* in both chromosomes (Figure 9A,C), including genome gaps and one inversion event in chromosome 1 (Figure 9A,B). 

The whole genome alignment identified 18 locally collinear blocks (LCBs) in *Vibrio* sp. J383, which are conserved segments with no genomic rearrangements [43]. The comparative alignment analysis of each chromosome showed 9 LCBs in chromosome 1 (Figure 9B) and 9 LCBs in chromosome 2 (Figure 9D). 

Also, comprehensive genome analysis in PATRIC indicated that there were some pathogenesis-associated genes. Chromosomes 1 and 2 contain several transporter-related genes as well as genes linked to virulence or antibiotic resistance. RAST and PATRIC comprehensive genome analyses identified the presence or absence of specific genes in the chromosomes and the plasmid (Appendix A). 

## 4. Discussion

The causes of ulcerative skin disease in Atlantic salmon are not fully understood. Several causative agents are being described in the North Atlantic rim, including *M. viscosa*, *Tenecebacillum* spp, and *A. wondandis* [4,5]. However, differences in disease etiology indicate that several undescribed pathogens might also cause skin ulcers in Atlantic salmon. In the present study, we isolated *Vibrio* sp. strain J383 from the spleen of Atlantic salmon exhibiting skin ulcers (Figure 1). The phenotypic characterization indicates that this strain is marine and requires the presence of at least 1% of NaCl for survival (Table 1 and Appendix A). Also, *Vibrio* sp. J383 showed substantial growth between 4 and 15 °C, but no growth at temperatures over 28 °C, suggesting that this strain is adapted to cold temperatures. *Vibrio* sp. J383 also possesses several virulence factors, including hemolysins, siderophores, and LPS (Table 1, Appendix A), indicating that it has pathogenic properties. Finally, *Vibrio* sp. J383 constitutively synthesizes siderophores, indicating that this strain can scavenge essential iron within and outside the host. The synthesis of siderophores is usually regulated [44], but *Vibrio* sp. J383 possesses a natural constitutive expression that requires further study. 

Infection assays indicated that *Vibrio* sp. J383 is a non-acute, chronic pathogen that can produce skin ulcers and kill fish, especially at 1 2 °C (Figure 2 and Figure 3). *Vibrio* sp. J383 triggered clinical signs of ulcer disease in vaccinated farmed Atlantic salmon, indicating that the generic vaccine utilized does not confer protection against this novel chronic pathogen (Figure 2B). Skin ulcer severity ranged from mild to severe in some of the fish that were infected with medium and high doses (10^7^ and 10^8^ CFU/dose, respectively) (Figure 2B). Although clinical signs of ulcer disease were evident in the infected fish, the low mortality rates indicated that this strain is not an acute pathogen. Around 5% to 22.5% mortality was recorded in the infected fish given the medium and high doses of *Vibrio* sp. J383 (Figure 2A). Our result is consistent with the literature, which indicates that the mortality during outbreaks in sea-cages with skin ulcers ranged from 0.01 to 23.32% [13]. *Vibrio* sp. J383 caused bacteremia, and it was detected until 10 wpi in Atlantic salmon at 12 °C. Bacteremia showed the same patterns in both infection assays (Figure 4A, Table 2). This indicates that *Vibrio* sp. J383 can cause a systemic infection. *Vibrio* sp. J383 was detected at 6 wpi in most of the blood samples in the high-dose infection group (Table 2), and isolated in all the blood samples at 12 °C (Figure 4A). Bacteremia was not recorded at 10 °C at 2 wpi but was reported at 12 °C in some blood samples (Table 2, Figure 4A). No bacteremia and mortality were recorded when the temperature increased to 16 °C (Table 2). These results are consistent with previous field observations, which indicated that the water temperature during skin ulcerative disease outbreaks ranged from 10 °C to 13 °C [13]. Our findings indicated that *Vibrio* sp. J383 becomes more invasive at 12 °C compared to 10 °C. Regardless of water temperature, *Vibrio* sp. J383 was not detected in the blood at 12 wpi (Table 2, Figure 4A). However, *Vibrio* sp. J383 was detected in the spleen, head kidney, and liver at 12 and 14 wpi in Atlantic salmon infected with the high dose at 12 °C (Figure 4B). Bacterial loads were significantly higher in the spleen compared to the liver and head kidney at 12 and 14 wpi. *Vibrio* sp. J383 loads decreased substantially from 12 wpi to 14 wpi (Figure 4B). Collectively, our findings suggest that *Vibrio* sp. J383 is a chronic pathogen. 

The biochemical characterization and identification of environmental *Vibrio* species have been complicated because of their notable diversity [45,46]. The biochemical profile attained using API 20NE showed that *Vibrio* sp. J383 was unable to reduce urea but reduces nitrates and produces indole, suggesting 64.8% and 34.6% similarity to *V. vulnificus* and *V. alginolyticus*, respectively (Appendix A). Phylogenetic and comparative analyses showed that *Vibrio* sp. J383 is closely related to *V. splendidus*, with 93% identity, and this suggests that these two strains share a common ancestor. However, phenotypical tests revealed significant differences between *Vibrio* sp. J383 and other *Vibrio* strains, indicating that *Vibrio* sp. J383 could be a novel species. 

*V. anguillarum* J360, a virulent strain isolated from the North Atlantic, displayed thermo-inducible α-hemolysin activity at 28 °C, but no hemolytic activity at 15 °C [15]. In contrast, *Vibrio* sp. J383 showed hemolysin activity at 15 °C but not at 28 °C. *Vibrio* spp. utilize hemolysins to lyse host erythrocytes to acquire nutrients, such as iron [47]. Hemolysins are crucial virulence factors for *V. anguillarum* and play a key role in boosting its pathogenicity [48,49]. The presence of hemolytic activity in *Vibrio* sp. J383 indicates its potential pathogenicity. *Vibrio* sp. J383 showed growth at 4 °C, optimal growth around 15 °C, weak growth at 28 °C and no growth at 37 °C. Also, *Vibrio* sp. J383 produces catalase and oxidase. These results are consistent with most pathogenic marine *Vibrio* spp. [50], and the ability of *Vibrio* sp. J383 to grow at 4 °C indicates that this novel pathogen is well adapted to cold environments.

The genotypic characterization of *Vibrio* sp. J383 indicated the presence of two chromosomes. The existence of two chromosomes is a basic characteristic of *Vibrio* spp. that developed as a survival means, and allows for the rapid adaptation of the pathogen to different environments and hosts [51]. *Vibrio* sp. J383 has a genome size of 5,902,734 bp, very similar to *V. splendidus* BST 398, (5,508,387 bp) (Table 3). The genome of *V. splendidus* BST 398 included 4700 predicted open reading frames, a G + C content of 44.12%, 137 tRNA genes and 46 rRNA genes [52]. In contrast, the whole genome of the novel strain *Vibrio* sp. J383 has a total of 16 (5S), 15 (16S), and 15 (23S) rRNAs, 138 tRNAs, and 5 ncRNAs (Table 4). These results suggest that although *V. splendidus* and *Vibrio* sp. J383 share a common ancestor, they are different strains with distinct genomic characteristics. 

Some specific genes associated with virulence and antibiotic resistance were detected in the PATRIC comprehensive genome analysis. The tetracycline resistance subsystem was found in *Vibrio* sp. J383, which makes this strain resistant to antibiotics and toxic compounds. Hydroxyacylglutathione hydrolase was also found in *Vibrio* sp. J383, which is a virulence-related gene and contributes to the stress response in bacteria.

Cold shock proteins of the CSP family were found in chromosome 2 of *Vibrio* sp. J383 (Appendix A). Cold shock proteins help cells adapt by reducing some of the negative effects of temperature changes [53]. They play a crucial role in the cold shock response, and recent data suggest that CSPs may have a greater role in bacterial stress tolerance [54,55,56]. Following the initial cold shock response, the production of CSPs declines while the production of other proteins increases. This helps the cells to grow at a low temperature, but at a slower rate [57]. This finding can explain why *Vibrio* sp. J383 has fast growth at 15 °C and slightly slower growth at 4 °C (Table 1).

Flagellum was detected in chromosome 1 of *Vibrio* sp. J383 (Appendix A), which plays a key role in bacteria motility, and can contribute to biofilm formation, protein export and adhesion [58]. Chemotaxis present in chromosome 1 of *Vibrio* sp. J383 plays several roles, including biofilm formation, auto aggregation and swarming, as well as in bacterial interactions with their hosts [59]. Toxin–antitoxin systems are common in bacterial genomes and found in chromosome 2 of *Vibrio* sp. J383 (Appendix A). They are normally made of two elements: a toxin that inhibits a vital cellular process and an antitoxin that hinders its cognate toxin [60]. 

A total of 24 genomic islands were detected in *Vibrio* sp. J383. Genomic Islands have also been identified in the species *V. anguillarum*. For example, *V. anguillarum* J360 has 21 GIs [15]. Genes encoding for integrase and transposase were found in the mentioned *Vibrio* strains. Glutaredoxin was detected in chromosome 2 of *Vibrio* sp. J383. and performs a critical role in the protection against oxidative stress in bacteria [61]. VirK is a virulence factor present in several bacterial pathogens that has been shown to contribute to *Salmonella enterica* serovar Typhimurium and *Escherichia coli* virulence [62,63,64]. It may contribute to *Vibrio* sp.’s pathogenicity as well. Also, Zot was detected in the genomes of *V. parahaemolyticus* [65], indicating a correlation between this gene and the cytotoxicity of bacteria [65,66]. In *V. cholerae*, Zot is an important toxin after the classical cholera toxin (CT), and it is encoded by the CTX prophage [67]. It has been shown that Zot has enterotoxic activity and it is hypothesized that it plays a role in the classic diarrhea symptom of *V. cholerae* infections [68,69]. The presence of Zot in the genome of *Vibrio* sp. J383 perhaps contributes to its virulence in fish hosts. 

The T6SS is a contractile nanomachine that plays a role in interbacterial competition and bacterial pathogenesis by secreting toxic effector proteins into adjacent cells [70,71,72,73,74,75]. The *Vibrio* sp. J383 genome includes two T6SSs on genomic islands 12 and 21 (Table 5, Figure 7, and Appendix A). T6SS apparatuses are composed of 13 core components that form two parts, the membrane complex and phage-like tail [76,77]. Three membrane-associated proteins comprise the membrane complex, TssJ, TssL, and TssM [75,76,78,79]. Consistent with this, the *Vibrio* sp. J383 homologs are predicted to be lipidated or to contain transmembrane domains. The T6SS tail is evolutionarily related to the contractile tail of the T4 bacteriophage and is composed of several subassemblies. The baseplate is composed of *tssE*, *tssF*, *tssG*, and *tssK* [80,81], and the *Vibrio* sp. J383 homologs share 60–80% identity with other *Vibrio* T6SS counterparts. *tssB*, *tssC*, and *tssA* comprise the sheath and cap of the T6SS and are encoded in both loci [82,83,84]. Of note, *tssB* and *tssC* share greater than 95% identity with other *Vibrio* T6SSs. Hcp and VgrG play a role in substrate recognition and are co-secreted with associated toxins, and both are encoded in the T6SS loci in *Vibrio* sp. J383 [76,78,85]. 

The genes that encode these T6SS apparatus subunits share high levels of homology between genomic islands 12 and 21. Based on homology with well-characterized T6SSs, both apparatuses encoded by *Vibrio* sp. J383 belong to the T6SS*^i^* family. It is impossible to identify T6SS-exported toxins based on sequence alone; however, some of the hypothetical genes in these two loci encode hallmarks that suggest that they may be effector proteins. The presence of toxic effectors genes in close proximity to T6SS apparatus genes does not preclude other effectors encoded in distant genomic loci, and further research can identify these toxins in the future.

Gene duplications are essential prerequisites for gene innovation, which may assist in adaptation to changing environmental conditions [86]. In chromosome 1 of *Vibrio* sp. J383, *tssC* is duplicated (Figure 7). T6SSs have been strongly linked to a variety of biological processes, including biofilm formation, bacterial survival in the environment, virulence, and host adaptation; therefore, the duplication of *tssC* in *Vibrio* sp. J383 may be an important step toward increasing the fitness of this strain in the environment, amongst the microbiota, or as a pathogen [87,88,89].

In summary, *Vibrio* sp. J383 has several virulence factors and genes that are associated with pathogens. 

*Vibrio* sp. J383 has one large plasmid, and an analysis based on PATRIC annotation found one subsystem, the MazEF toxin–antitoxin (program cell death) system (Appendix A). Toxin–antitoxin (TA) systems, initially discovered in plasmids, were recognized as extra-chromosomal genes responsible for post-segregationally killing, which protects plasmid integrity [90]. Toxin–antitoxin (TA) systems have been reported in many bacterial genomes and mediate program cell death (PCD), and are therefore attractive targets for new anti-microbial drugs since they are recognized to “kill from within” [91,92]. The antitoxins neutralize the toxin using different mechanisms and play vital roles, including the maintenance of genomic stability, and assist in biofilm formation in some bacteria [90]. The presence of the TA system in the plasmid of *Vibrio* sp. J383 indicates that it may play may the same role and enable *Vibrio* sp. J383 to survive in different temperatures and maintain its virulence factors.

## 5. Conclusions

The biochemical profile showed that *Vibrio* sp. J383 is similar to *V. vulnificus*, with 64.8% identity. However, phylogenetic and comparative analyses showed that *Vibrio* sp. J383 is closely related to *V. splendidus*, with 93% identity. The isolation of *Vibrio* sp. J383 from blood samples confirmed Koch’s postulates. Mortality was approximately 20% in the vaccinated fish infected with a 10^8^ CFU/dose, but no mortality was observed in fish infected with a low dose (10^6^ CFU/dose). *Vibrio* sp. J383 was detected for 10 wpi in the blood and at up to 14 wpi in the internal organs (spleen and kidney). The pathogenicity of this new strain is supported by Koch’s postulates and the presence of pathogenic genomic islands (GIs 12 and 21) containing virulence factors such as type VI secretion system (T6SS) genes and multidrug-resistance transporter/family protein. *Vibrio* sp. J383 displays unique characteristics, and has notable differences, compared to other *Vibrio* strains. The results of this study revealed that *Vibrio* sp. J383 is potentially a new species that can trigger clinical signs of ulcer disease and cause chronic infections in vaccinated farmed Atlantic salmon. The impact of the co-infection of *Vibrio* sp. J383 with other etiological agents, like *Moritella viscosa*, on Atlantic salmon remains to be investigated.

## Figures and Tables

**Figure 1 microorganisms-11-01736-f001:**
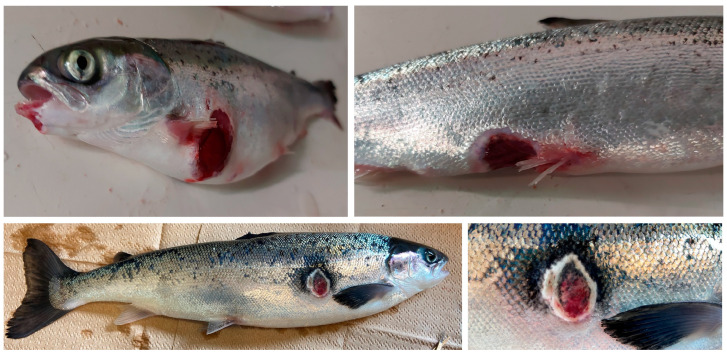
*Vibrio* sp. J383 strain was isolated from the spleen of Atlantic salmon exhibiting clinical signs of skin ulcers.

**Figure 2 microorganisms-11-01736-f002:**
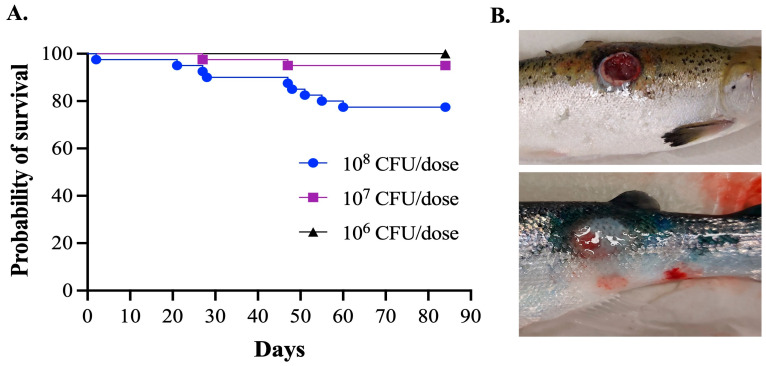
Survival and clinical signs of Atlantic salmon interperitoneally infected with *Vibrio* sp.: (**A**) Mortality of Atlantic salmon infected with *Vibrio* sp. J383; and (**B**) clinical sign of Atlantic salmon infected with *Vibrio* sp. J383.

**Figure 3 microorganisms-11-01736-f003:**
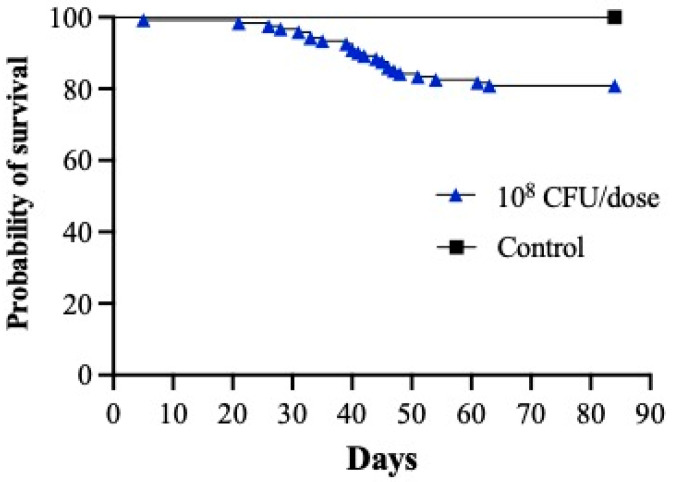
Survival portion of Atlantic salmon interperitoneally infected with *Vibrio* sp. J383 at a high dose at 12 °C.

**Figure 4 microorganisms-11-01736-f004:**
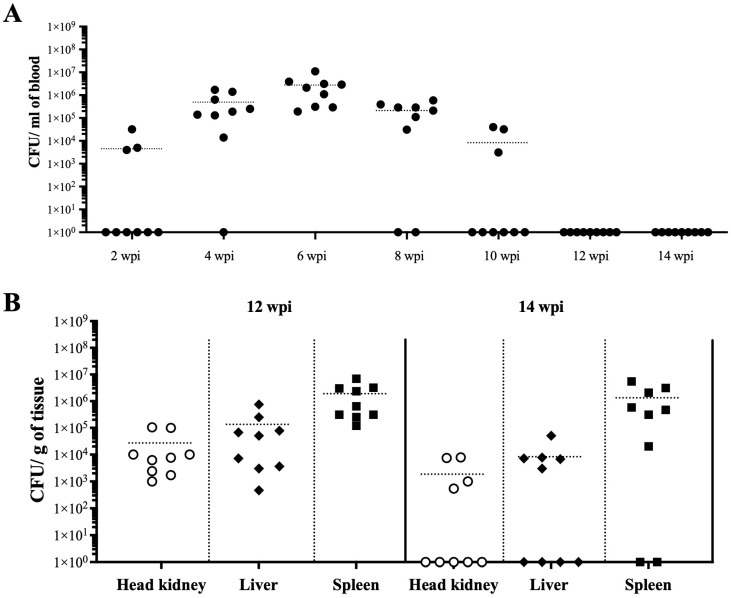
*Vibrio* sp. J383 blood and tissue colonization in vaccinated farmed Atlantic salmon. (**A**) *Vibrio* sp. J383 loads in blood of (*n* = 9) infected fish with the high dose (1 × 10^8^ CFU/dose) at 2, 4, 4, 6, 8, 10, 12 and 14 wpi; and (**B**) *Vibrio* sp. J383 loads in head kidney, liver, spleen of (*n* = 9) infected fish with the high dose (1 × 10^8^ CFU/dose) of *Vibrio* sp. J383 at 12 and 14 wpi. Full circle: blood; empty circle: head kidney; full roboid: liver; full square: spleen.

**Figure 5 microorganisms-11-01736-f005:**
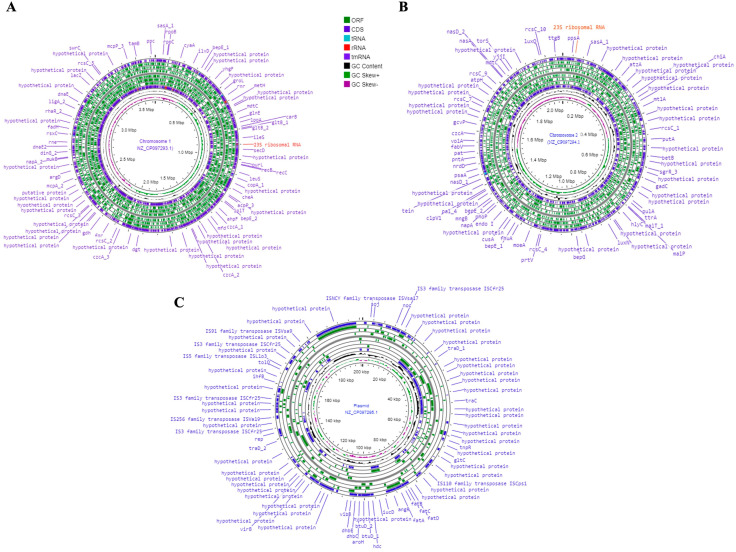
*Vibrio*. sp. J383 chromosomes. (**A**) *Vibrio*. sp. J383 chromosome 1 genome visualization; (**B**) *Vibrio*. sp. J383 chromosome 2 genome visualization; and (**C**) genome map representation of the large plasmid of *Vibrio* sp. J383. A circular graphical display of the distribution of the genome annotations is provided. This includes, from outer to inner rings, the contigs, CDS on the forward strand, CDS on the reverse strand, RNA genes, CDS with homology to known antimicrobial resistance genes, CDS with homology to known virulence factors, GC content and GC skew.

**Figure 6 microorganisms-11-01736-f006:**
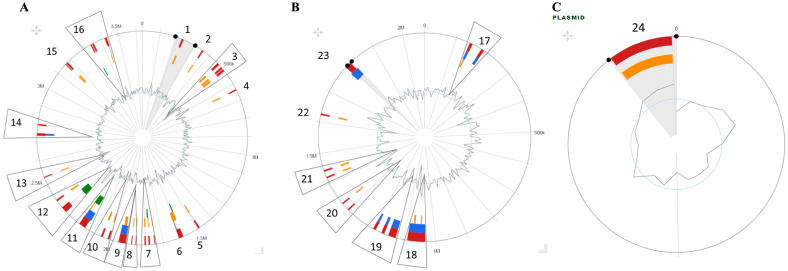
Genomic islands (GIs) detected in *Vibrio* sp. J383. (**A**) Chromosome 1 and (**B**) chromosome 2; and (**C**) genomic islands (GIs) detected in plasmid. Red bars represent GIs detected using 3 different packages; blue bars represent the GIs detected with SIGI-HMM package; orange bars represent the GIs detected with the Island Path-DIMOB package; green bars represent the GIs detected with the Island Pick package.

**Figure 7 microorganisms-11-01736-f007:**
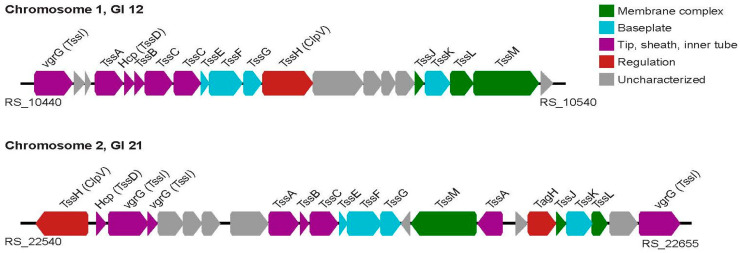
Type VI secretion system (T6SS) gene cluster in *Vibrio* sp. J383. The T6SS in chromosomes 1 and 2 are made from 13 Tss (Type Six Subunits) proteins that are known as the “core components”. TssC gene duplicated in chromosome 1. Several uncharacterized genes are present in two T6SS loci and these may encode toxins secreted by the apparatus; however, further characterization is required to elucidate the role of these genes.

**Figure 8 microorganisms-11-01736-f008:**
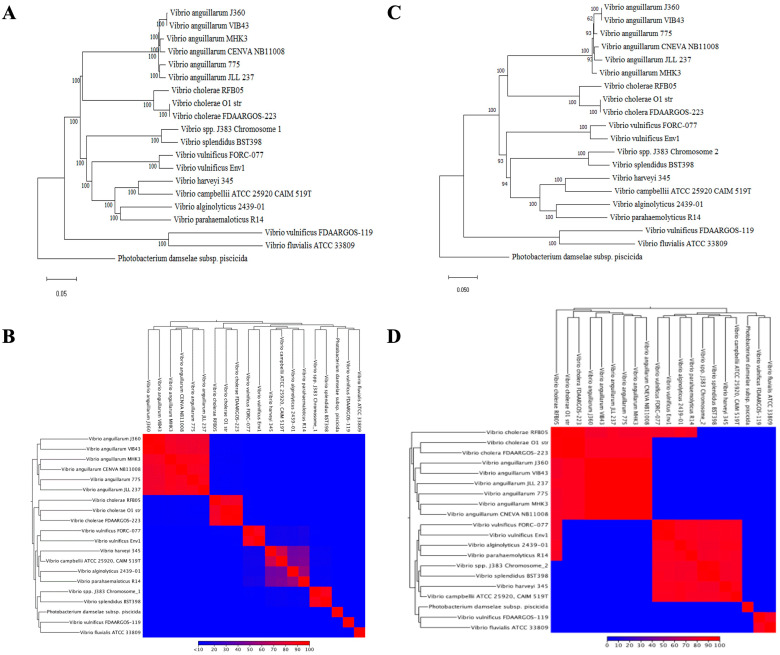
Phylogenetic history of *Vibrio* sp. J383 genome. (**A**) Chromosome 1 evolutionary history was inferred using the neighbor-joining method, with a bootstrap consensus of 500 replicates for taxa analysis in MEGA 11 software; (**B**) heat map visualization of aligned sequence’s identities for *Vibrio* sp. J383 chromosome 1; genome alignment involved 20 *Vibrio* sp, analysis in CLC; and (**C**) chromosome 2 evolutionary history was inferred using the neighbor-joining method, with a bootstrap consensus of 500 replicates for taxa analysis in MEGA 11 software. (**D**) Heat map visualization of aligned sequences identified for *Vibrio* sp. J383 chromosome 2 genome alignment. This involved 20 *Vibrio* sp, and analysis in CLC.

**Figure 9 microorganisms-11-01736-f009:**
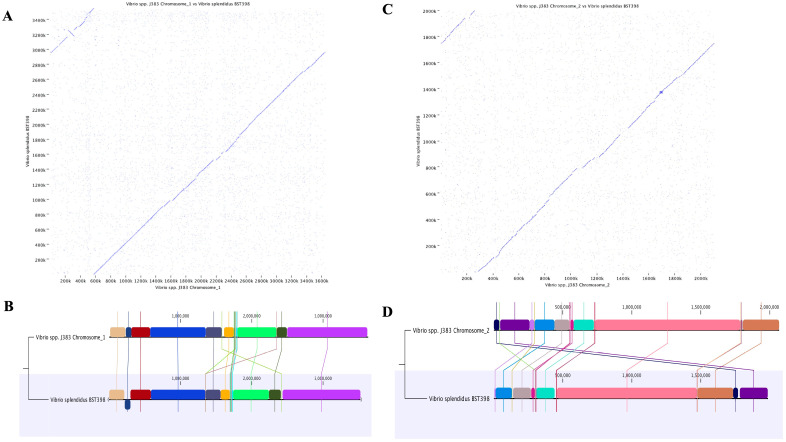
Comparative genome synteny between *Vibrio* sp. J383 and *V. splendidus* BST 398. (**A**) Dot plot analysis for chromosome 1; dot plots were computed using CLC Genomics Workbench v.20; blue arrow represents inversion. (**B**) Homologous regions identified as locally colinear blocks (LCBs) of chromosome 1. (**C**) Dot plot analysis for chromosome 2; Dot plots were computed using CLC Genomics Workbench v.20. (**D**) Homologous regions identified as locally colinear blocks of chromosome 2.

**Table 1 microorganisms-11-01736-t001:** Phenotypic characteristics of *Vibrio* sp. J383.

Characteristics (Growth at)	*Vibrio* J383
Gram Stain	Gram-Negative
Capsule stain	+
Hemolysin in Salmon blood agar(15 °C)	+
Hemolysin in Sheep blood agar (15 °C)	+
Hemolysin in Salmon blood agar(28 °C)	−
Hemolysin in Sheep blood agar (28 °C)	−
Type 1 fimbria	−
Growing in LB 0% NaCl (15 °C)	−
Growing in LB 0.5% NaCl (15 °C)	−
Growing in TSB 2% NaCl (4 °C)	+
Growing in TSB 2% NaCl (15 °C)	+
Growing in TSB 2% NaCl (28 °C)	+
Growing in TSB 2% NaCl (37 °C)	−
Motility Test	+
Catalase	+
Oxidase	+
Biofilm	+
**Antibiogram using sensi-disk of:**	**Halo diameter (mm)**
Vibriostatic agent (O-129)	25 (Susceptible)
Tetracycline (10 μg)	30 (Susceptible)
Oxytetracycline (30 μg)	30 (Susceptible)
Ampicillin (10 μg)	23 (Susceptible)
Sulfamethoxazole (25 μg)	25 (Susceptible)
Chloramphenicol (30 μg)	30 (Susceptible)
Colistin sulphate (10 μg)	0 (Resistant)
Oxalinic acid (2 μg)	24 (Susceptible)

**Table 2 microorganisms-11-01736-t002:** *Vibrio* sp. J383 isolated from blood samples at different time points post-infection (weeks post-infection: wpi).

Positive Samples for *Vibrio* spp. J383 (Total Positive/6 Fish)						
Temperature	10 °C		12 °C		16 °C	
Dose	2 wpi	4 wpi	6 wpi	8 wpi	10 wpi	12 wpi
10^6^	0/6	0/6	0/6	0/6	0/6	0/6
10^7^	0/6	0/6	1/6	0/6	1/6	0/6
10^8^	0/6	3/6	4/6	3/6	2/6	0/6

**Table 3 microorganisms-11-01736-t003:** Rapid annotation subsystem technology (RAST) *Vibrio* sp. J383 annotation.

Characteristics	Chromosome-1	Chromosome-2	Plasmid
Genome size (bp)	3,633,265	2,068,312	201,166
G + C content (%)	44.3	44.1	43.4
Number of subsystems	309	101	8
Number of coding sequences	3235	1866	237
Number of RNAs	163	21	0

**Table 4 microorganisms-11-01736-t004:** Prokaryotic genome annotation summary (*Vibrio* sp. J383).

Attribute	Data Provider
Annotation Pipeline	NCBI prokaryotic Genome Annotation pipeline
Annotation Method	Best-placed reference protein set; GeneMarkS-2+
Genes (total)	5288
CDSs (total)	5099
Genes (coding)	5031
CDSs (with protein)	5031
Genes (RNA)	189
rRNAs	16, 15, 15 (5S, 16S, 23S)
Complete rRNAs	16, 15, 15 (5S, 16S, 23S)
tRNAs	138
ncRNAs	5
Pseudo Genes (total)	68
CDSs (without protein)	68
Pseudo Genes (ambiguous residues)	0 of 68
Pseudo Genes (frameshifted)	29 of 68
Pseudo Genes (incomplete)	33 of 68
Pseudo Genes (internal stop)	21 of 68
Pseudo Genes (multiple problems)	13 of 68

**Table 5 microorganisms-11-01736-t005:** Genes of type VI secretion system (T6SS) in *Vibrio* sp. J383.

Gene	Locus Tag	Chromosome/GI	Location(nt)		Putative Function
*vgrG*	M4S28_RS10440	1/12	2,282,509	2,284,491	Tip of the T6SS apparatus
	M4S28_RS10445	1/12	2,284,552	2,285,112	Unknown
	M4S28_RS10450	1/12	2,285,122	2,285,421	Unknown
*tssA*	M4S28_RS10455	1/12	2,285,641	2,287,107	Cap of the T6SS sheath
*hcp*	M4S28_RS10460	1/12	2,287,140	2,287,661	Inner tube of the T6SS
*tssB*	M4S28_RS10465	1/12	2,287,681	2,288,184	T6SS sheath
*tssC*	M4S28_RS10470	1/12	2,288,184	2,289,662	T6SS sheath
*tssC*	M4S28_RS10475	1/12	2,289,701	2,291,095	T6SS sheath
*tssE*	M4S28_RS10480	1/12	2,291,095	2,291,517	T6SS baseplate
*tssF*	M4S28_RS10485	1/12	2,291,510	2,293,261	T6SS baseplate
*tssG*	M4S28_RS10490	1/12	2,293,326	2,294,258	T6SS baseplate
*tssH*	M4S28_RS10495	1/12	2,294,306	2,296,918	Disassembly of the T6SS apparatus
	M4S28_RS10500	1/12	2,296,928	2,299,495	MFS transporter
	M4S28_RS10505	1/12	2,299,492	2,300,490	ABC transporter protein
	M4S28_RS10510	1/12	2,300,477	2,301,205	Transporter protein
	M4S28_RS10515	1/12	2,301,209	2,302,159	FHA domain-containing protein
*tssJ*	M4S28_RS10520	1/12	2,302,156	2,302,644	T6SS membrane complex
*tssK*	M4S28_RS10525	1/12	2,302,686	2,304,017	T6SS baseplate
*tssL*	M4S28_RS10530	1/12	2,304,023	2,305,219	T6SS membrane complex
*tssM*	M4S28_RS10535	1/12	2,305,222	2,308,614	T6SS membrane complex
	M4S28_RS10540	1/12	2,308,694	2,309,347	AarF/UbiB family protein
*tssH*	M4S28_RS22540	2/21	1,384,843	1,387,536	Disassembly of the T6SS apparatus
*hcp*	M4S28_RS22545	2/21	1,387,991	1,388,509	Inner tube of the T6SS
*vgrG*	M4S28_RS22550	2/21	1,388,584	1,390,662	Tip of the T6SS apparatus
*vgrG*	M4S28_RS22555	2/21	1,390,662	1,391,147	Tip of the T6SS apparatus
	M4S28_RS22560	2/21	1,391,173	1,392,492	Unknown
	M4S28_RS22565	2/21	1,392,473	1,393,465	Unknown
	M4S28_RS22570	2/21	1,393,458	1,394,411	Unknown
	M4S28_RS22575	2/21	1,394,912	1,396,903	Unknown
*tssA*	M4S28_RS22580	2/21	1,396,905	1,398,479	Cap of the T6SS sheath
*tssB*	M4S28_RS22585	2/21	1,398,497	1,399,003	T6SS sheath
*tssC*	M4S28_RS22590	2/21	1,399,012	1,400,487	T6SS sheath
*tssE*	M4S28_RS22595	2/21	1,400,548	1,400,958	T6SS baseplate
*tssF*	M4S28_RS22600	2/21	1,400,969	1,402,717	T6SS baseplate
*tssG*	M4S28_RS22605	2/21	1,402,714	1,403,712	T6SS baseplate
	M4S28_RS22610	2/21	1,403,742	1,404,194	Lrp/AsnC transcriptional regulator
*tssM*	M4S28_RS22615	2/21	1,404,255	1,407,647	T6SS membrane complex
*tssA*	M4S28_RS22620	2/21	1,407,701	1,409,011	Cap of the T6SS sheath
	M4S28_RS22625	2/21	1,409,679	1,410,284	Unknown
*tagH*	M4S28_RS22630	2/21	1,410,294	1,411,790	Regulatory
*tssJ*	M4S28_RS22635	2/21	1,411,783	1,412,283	T6SS membrane complex
*tssK*	M4S28_RS22640	2/21	1,412,295	1,413,620	T6SS baseplate
*tssL*	M4S28_RS22645	2/21	1,413,617	1,414,408	T6SS membrane complex
	M4S28_RS22650	2/21	1,414,557	1,416,014	Unknown
*vgrG*	M4S28_RS22655	2/21	1,416,026	1,418,182	Tip of the T6SS apparatus

## Data Availability

Not applicable.

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
