# Peer review of "Comparative Genomic Analysis of a Novel *Vibrio* sp. Isolated from an Ulcer Disease Event in Atlantic Salmon (*Salmo salar*)"

_microorganisms, 2023, doi:10.3390/microorganisms11071736_

Round 1
Reviewer 1 Report
The authors isolated cold-growing Vibrio spp. from Atlantic salmon developing skin ulcers and performed characterization, pathogenicity testing, and whole genome analysis. It showed that the isolated bacterium was a novel Vibrio sp. and involved in salmon pathogenicity. It is also novel as a bacterium associated with salmon pathogenicity and as a Vibrio species, and is an important finding in the elucidation of salmon skin ulcers caused by newly discovered bacteria. Analysis of the ecology and pathogenicity of this novel Vibrio sp. will be important for advancing research on the pathogenesis of salmon skin ulcers. Also, look up the biological and genetical characteristics of each; the results of the J383 infection experiment describe that J383 was detected in all three tissues of the above mentioned.
In 2.1.1. Isolation, the spleen, head kidney, and liver were used as materials to try to isolate the bacteria, but the description in Figure 1 stated that it was from the spleen. The results in 3.1. did not include a description of what kind of bacteria were isolated from which materials and how many. Was only J383 isolated? If multiple colonies are detected, they can be separated using hemolytic differences as an indicator when spread on blood agar medium. That would be very important to know the relationship to skin ulcers. Please provide details.
In 2.1.2., to show the properties of J383, it would be good to plot the incubation temperature and NaCl concentration on the horizontal axis and the highest value of bacterial growth (O.D.) on the vertical axis.
In 2.2.3. authors were conducting an infection experiment with a fairly high number of bacteria. As cited by the authors, ref#4 (J Fish Dis. 1998 Jan;21(1):19-28.), the LD50 for V. marinus was 3.5 x 10^3. If this is used as a positive control and a comparison is made using another environmental isolate as a negative control, the pathogenic relevance would be apparent.
In 3.5., T6SS was encoded in GI12 and GI21, respectively, please compare their mutual homology (e.g. vgrG, etc.) and show the data. Also describe in 4. Discussion any functional differences as seen in T3SS1, T3SS2 in V. parahaemolyticus (Microbiol Immunol. 2020 Mar;64(3):167-181.).
In 4. Discussion, L399-L425, was there any correlation between mortality and the disappearance of the bacteria in vivo or the persistence of ulcers? Since excretion of the bacteria from the fish would induce a secondary infection, was the bacteria excreted into the environment from the infected salmon? Was there a correlation between the emergence and disappearance of skin ulcers and the disappearance of the bacteria in the various tissues?
L437, . .hemolysin activity at 15 C but not at 28 C. Wasn't it dependent on bacterial growth (log phase or plateau)?
L393-L394, L442-L443, virulence factors, catalase and oxidase. Catalase and oxidase are basic characteristics of Vibrio spp. and are not considered virulence factors.
L461-L468, authors mentioned CSP, but have you considered heat shock protein (HSP) inhibition of growth at 28 C<?
P. 1, L40, Vibrio wodanins -> Vibrio wodanis
P. 1, L43, A wondanis -> A. wodanis
P. 15, L449, 5,902,734 Kb -> 5,902,734 bp
P. 18, L574, No reference name -> J Fish Dis. 1998 Jan;21(1):19-28.
Author Response
Dear reviewers,
We thank you very much for the comments and suggestions. We have made revisions according to the referees’ comments as described below.
Reviewer 1
The authors isolated cold-growing Vibrio spp. from Atlantic salmon developing skin ulcers and performed characterization, pathogenicity testing, and whole genome analysis. It showed that the isolated bacterium was a novel Vibrio sp. and involved in salmon pathogenicity. It is also novel as a bacterium associated with salmon pathogenicity and as a Vibrio species and is an important finding in the elucidation of salmon skin ulcers caused by newly discovered bacteria. Analysis of the ecology and pathogenicity of this novel Vibrio sp. will be important for advancing research on the pathogenesis of salmon skin ulcers. Also, look up the biological and genetical characteristics of each; the results of the J383 infection experiment describe that J383 was detected in all three tissues of the above mentioned.
In 2.1.1. Isolation, the spleen, head kidney, and liver were used as materials to try to isolate the bacteria, but the description in Figure 1 stated that it was from the spleen. The results in 3.1. did not include a description of what kind of bacteria were isolated from which materials and how many. Was only J383 isolated? If multiple colonies are detected, they can be separated using hemolytic differences as an indicator when spread on blood agar medium. That would be very important to know the relationship to skin ulcers. Please provide details.
Response (RE):
We added the details to the supplementary information as follows:
“We isolated a total of five strains, 3 from the head kidney, 1 from the liver, and 1 from the spleen, from different infected fish. An initial screening to determine the virulence of the five isolates was conducted in Atlantic salmon (200 g). The fish were transferred to the AQ2/3 biocontainment zone of the CDRF, acclimated for 1 week, and intraperitoneally (ip) injected with a very high dose (1x108 CFU/dose) of each isolate. Each treatment consisted of 6 fish in individual tanks under optimal conditions. Fish ip injected with PBS were used as a negative control, and fish ip injected with M. viscosa J311 were used as a positive control. As expected, M. viscosa killed all the animals quickly, likely due to ‘toxic shock’ (the release of toxins by the bacteria), and all the animals ip injected with PBS survived (Figure S2). Only strain J383 (SP6) caused mortality and clinical signs of ‘ulcer disease’. However, the mortality associated with J383 infection in Atlantic salmon indicates that it causes a chronic type of infection rather than an acute infection. This is consistent with the infection event from which these samples were obtained.
Figure S2. Acute mortality of Atlantic salmon infected with different bacterial strains isolated from fish exhibiting ‘winter ulcer’-like clinical signs. Moritella viscosa J311 was obtained from American Type Culture Collection, NHI. The animals were injected with 100 ul of pathogen (108 CFU / dose) or PBS.”
The main text was also modified to include this additional information:
Lines 151-157 “The infection procedures were conducted in the AQ2 biocontainment unit at the Cold-Ocean Deep-Sea Research Facility (CDRF), MUN. Fish were transferred to the AQ2-CDRF unit and acclimated for one week at 10°C before infection. An initial infection screening assay was conducted in Atlantic salmon (200 g) intraperitoneally (ip) injected with a high dose (1x108 CFU/dose) of each isolated strain. Each infected group consisted of 5 fish in individual tanks under optimal conditions. Fish ip injected with PBS were used as a negative control, and fish ip injected with M. viscosa J311 (ATCC BAA-105) were used as a positive control.”
Lines 315-326: “We isolated a total of five strains, 3 from the head kidney, 1 from the liver, and 1 from the spleen, from different infected fish. An initial screening to determine the virulence of the five isolates was conducted in Atlantic salmon (200 g). The fish were transferred to the AQ2/3 biocontainment zone of the CDRF, acclimated for 1 week, and intraperitoneally (ip) injected with a very high dose (1x108 CFU/dose) of each isolate. Fish ip injected with PBS were used as a negative control, and fish ip injected with M. viscosa J311 were used as a positive control. As expected, M. viscosa killed all the animals quickly, and all the animals ip injected with PBS survived (Figure S2). Only the strain J383 (SP6) caused mortality and clinical signs of ulcer disease. However, the mortality associated with J383 infection in Atlantic salmon indicates that it causes a chronic type of infection rather than an acute infection. This is consistent with the infection event from which these samples were obtained.”
In 2.1.2., to show the properties of J383, it would be good to plot the incubation temperature and NaCl concentration on the horizontal axis and the highest value of bacterial growth (O.D.) on the vertical axis.
RE: J383 strain had the highest value of OD at 15°C. Figure S1J in the supplementary figures represents growth of this strain in TSB with 2% NaCl at 15°C. Vertical axis showed the (O.D.).
The suggestion from the reviewer implies to conduct new assays for a bacterial physiology type of study. We will consider this assay for future studies.
In 2.2.3. authors were conducting an infection experiment with a fairly high number of bacteria. As cited by the authors, ref#4 (J Fish Dis. 1998 Jan;21(1):19-28.), the LD50 for V. marinus was 3.5 x 10^3. If this is used as a positive control and a comparison is made using another environmental isolate as a negative control, the pathogenic relevance would be apparent.
RE: Please, see Figure S2. In the screening assay M. viscosa was used as positive control and PBS as negative control. We observed 100% mortality in positive control tank, but no mortality was recorded in negative control as expected.
In 3.5., T6SS was encoded in GI12 and GI21, respectively, please compare their mutual homology (e.g. vgrG, etc.) and show the data. Also describe in 4. Discussion any functional differences as seen in T3SS1, T3SS2 in V. parahaemolyticus (Microbiol Immunol. 2020 Mar;64(3):167-181.).
RE: We thank the reviewer for this comment. The sequences of the genes that encode subunits of the T6SS apparatus share high levels of homology with each other and also with well-characterized T6SSs. Regarding functional differences in potential effectors, it is impossible to compare differences without knowing their identities or functions. Our analysis suggests that some of the uncharacterized genes in these loci may encode secreted effectors; however, significant additional research would have to be performed to validate those hypotheses and this is beyond the scope of this manuscript. Furthermore, although T6SS effectors are sometimes encoded in close proximity to the apparatus genes, myriad examples of distantly encoded effector genes exist. We have included references to these points in the discussion and hope that these additions have improved the manuscript.
Please consider supplementary file 5 (comparison GI 12 and GI 21) that is added per your request in the supplementary data.
In 4. Discussion, L399-L425, was there any correlation between mortality and the disappearance of the bacteria in vivo or the persistence of ulcers? Since excretion of the bacteria from the fish would induce a secondary infection,
Was there a correlation between the emergence and disappearance of skin ulcers and the disappearance of the bacteria in the various tissues?
RE: There was a positive relation between appearance of clinical signs and the mortality in animals. J383 was reisolated in all the collected samples at 6 weeks post infection (Figure 4A) and the most mortality and clinical signs was recorded between 40 to 60 days (6 weeks to 8 weeks) post-infection (Figures 2A, 3). Also, no mortality was recorded after 9 weeks post infection (Figures 2A, 3) and the bacterial load in the blood substantially decreased from 8 to 10 weeks post infection (Figure 4A). No mortality clinical signs and bacteriemia observed after 12 wpi.
hemolysin activity at 15 C but not at 28 C. Wasn't it dependent on bacterial growth (log phase or plateau)?
RE: We did not observe any hemolysin activity before 24 hours, between 24 to 48 hours and even after 48 hours till 7 days incubation at 28ºC, so it seems that this Vibrio strain does not have any hemolysin activity in different time points at 28ºC and it is not related to the growth phase of the bacterium.
L393-L394, L442-L443, virulence factors, catalase and oxidase. Catalase and oxidase are basic characteristics of Vibrio spp. and are not considered virulence factors.
Reviewer is right. This sentence was modified to: Lines 672-677: “Vibrio sp. J383 also possesses several virulence factors, including hemolysins, siderophores, and lipases (Table 1, Figure S1E, F, G, I), indicating that it has pathogenic properties.”
L461-L468, authors mentioned CSP, but have you considered heat shock protein (HSP) inhibition of growth at 28 C<?
RE: Thanks for the suggestion. HSP should play a role in the adaptation and response to the temperature. We will consider this for further studies physiological studies.
- 1, L40, Vibrio wodanins -> Vibrio wodanis
- 1, L43, A wondanis -> A. wodanis
- 15, L449, 5,902,734 Kb -> 5,902,734 bp
- 18, L574, No reference name -> J Fish Dis. 1998 Jan;21(1):19-28.
RE: The sentences were corrected.

Reviewer 2 Report
1- Pls mention the number of naturally infected fish used for bacterial isolation
2- Number of the recovered isolates during the initial bacterial isolation
3- authors should highlight on the first isolation of V. splendidus as a potential pathogen
Author Response
REVIEWER 2
- Pls mention the number of naturally infected fish used for bacterial isolation
RE: All original strains were isolated from natural infection. After the bacterial strains were isolated, in this study we concentrate in the identification of the pathogen.
- Number of the recovered isolates during the initial bacterial isolation
We added the details to the supplementary information as follows:
“We isolated a total of five strains, 3 from the head kidney, 1 from the liver, and 1 from the spleen, from different infected fish. An initial screening to determine the virulence of the five isolates was conducted in Atlantic salmon (200 g). The fish were transferred to the AQ2/3 biocontainment zone of the CDRF, acclimated for 1 week, and intraperitoneally (ip) injected with a very high dose (1x108 CFU/dose) of each isolate. Each treatment consisted of 6 fish in individual tanks under optimal conditions. Fish ip injected with PBS were used as a negative control, and fish ip injected with M. viscosa J311 were used as a positive control. As expected, M. viscosa killed all the animals quickly, likely due to ‘toxic shock’ (the release of toxins by the bacteria), and all the animals ip injected with PBS survived (Figure S2). Only the strain J383 (SP6) caused mortality and clinical signs of ‘ulcer disease’. However, the mortality associated with J383 infection in Atlantic salmon indicates that it causes a chronic type of infection rather than an acute infection. This is consistent with the infection event from which these samples were obtained.
Figure S2. Acute mortality of Atlantic salmon infected with different bacterial strains isolated from fish exhibiting ‘winter ulcer’-like clinical signs. Moritella viscosa J311 was obtained from American Type Culture Collection, NHI. The animals were injected with 100 ul of pathogen (108 CFU / dose) or PBS.”
The main text was also modified to include this additional information:
Lines 151-157 “The infection procedures were conducted in the AQ2 biocontainment unit at the Cold-Ocean Deep-Sea Research Facility (CDRF), MUN. Fish were transferred to the AQ2-CDRF unit and acclimated for one week at 10°C before infection. An initial infection screening assay was conducted in Atlantic salmon (200 g) intraperitoneally (ip) injected with a high dose (1x108 CFU/dose) of each isolated strain. Each infected group consisted of 5 fish in individual tanks under optimal conditions. Fish ip injected with PBS were used as a negative control, and fish ip injected with M. viscosa J311 (ATCC BAA-105) were used as a positive control.”
Lines 315-326: “We isolated a total of five strains, 3 from the head kidney, 1 from the liver, and 1 from the spleen, from different infected fish. An initial screening to determine the virulence of the five isolates was conducted in Atlantic salmon (200 g). The fish were transferred to the AQ2/3 biocontainment zone of the CDRF, acclimated for 1 week, and intraperitoneally (ip) injected with a very high dose (1x108 CFU/dose) of each isolate. Fish ip injected with PBS were used as a negative control, and fish ip injected with M. viscosa J311 were used as a positive control. As expected, M. viscosa killed all the animals quickly, and all the animals ip injected with PBS survived (Figure S2). Only the strain J383 (SP6) caused mortality and clinical signs of ulcer disease. However, the mortality associated with J383 infection in Atlantic salmon indicates that it causes a chronic type of infection rather than an acute infection. This is consistent with the infection event from which these samples were obtained.”
3- authors should highlight on the first isolation of V. splendidus as a potential pathogen
RE: Trough the article we have indicated that Vibrio sp J383 is related to V. splendidus, with 93% identity. Line 305-306 indicate “The API 20NE profile 7474004 indicated that the isolate could be V. vulnificus or V. alginolyticus with 64.8% and 34.6% confidence, respectively (Table S1). However, we never identified J383 as V. splendidus.
